# Establishing need and population priorities to improve the health of homeless and vulnerably housed women, youth, and men: A Delphi consensus study

Esther S. Shoemaker[1,2,3,4], Claire E. Kendall[1,2,3,4,5], Christine Mathew[1], Sarah Crispo[1], Vivian Welch[1,6], Anne Andermann[7,8,9], Sebastian Mott[7], Christine Lalonde[1], Gary Bloch[10,11,12], Alain Mayhew[1], Tim Aubry[13], Peter Tugwell[1,2,4], Vicky Stergiopoulos[11,14], Kevin Pottie[1,2]*

1 Bruyère Research Institute, Ottawa, ON, Canada, 2 Department of Family Medicine, University of Ottawa, Ottawa, ON, Canada, 3 Institute of Clinical and Evaluative Sciences, Toronto, ON, Canada, 4 Ottawa Hospital Research Institute, Ottawa, ON, Canada, 5 Institute du Savoir Montfort, Ottawa, ON, Canada, 6 School of Epidemiology and Public Health, Faculty of Medicine, University of Ottawa, Ottawa, ON, Canada, 7 Department of Family Medicine, McGill University, Montreal, QC, Canada, 8 Department of Epidemiology, Biostatistics and Occupational Health, McGill University, Montreal, QC, Canada, 9 St Mary's Research Centre, St Mary's Hospital, Montreal, QC, Canada, 10 Department of Family and Community Medicine, St. Michael's Hospital, Toronto, ON, Canada, 11 Faculty of Medicine, University of Toronto, Toronto, ON, Canada, 12 Inner City Health Associates, Toronto, ON, Canada, 13 School of Psychology and Centre for Research on Educational and Community Services, University of Ottawa, Ottawa, ON, Canada, 14 Centre for Addictions and Mental Health, Toronto, ON, Canada

* kpottie@uottawa.ca

**Data Availability Statement:** All relevant data are within the paper and its Supporting Information files.

## Abstract

### Background

Homelessness is one of the most disabling and precarious living conditions. The objective of this Delphi consensus study was to identify priority needs and at-risk population sub-groups among homeless and vulnerably housed people to guide the development of a more responsive and person-centred clinical practice guideline.

### Methods

We used a literature review and expert working group to produce an initial list of needs and at-risk subgroups of homeless and vulnerably housed populations. We then followed a modified Delphi consensus method, asking expert health professionals, using electronic surveys, and persons with lived experience of homelessness, using oral surveys, to prioritize needs and at-risk sub-populations across Canada. Criteria for ranking included potential for impact, extent of inequities and burden of illness. We set ratings of $\geq 60\%$ to determine consensus over three rounds of surveys.

### Findings

Eighty four health professionals and 76 persons with lived experience of homelessness participated from across Canada, achieving an overall 73% response rate. The participants

**Funding:** This manuscript was funded by Inner City Health Associated (ICHA), Toronto, Canada. The funders had no role in study design, data collection and analysis, decision to publish, or preparation of the manuscript.

**Competing interests:** The authors have declared that no competing interests exist.

identified priority needs including mental health and addiction care, facilitating access to permanent housing, facilitating access to income support and case management/care coordination. Participants also ranked specific homeless sub-populations in need of additional research including: Indigenous Peoples (First Nations, Métis, and Inuit); youth, women and families; people with acquired brain injury, intellectual or physical disabilities; and refugees and other migrants.

## Interpretation

The inclusion of the perspectives of both expert health professionals and people with lived experience of homelessness provided validity in identifying real-world needs to guide systematic reviews in four key areas according to priority needs, as well as launch a number of working groups to explore how to adapt interventions for specific at-risk populations, to create evidence-based guidelines.

## Introduction

Homelessness is recognized as one of the most disabling and precarious conditions in high income countries [1]. Homelessness may be defined as a state in which an individual or family is without stable, permanent, or appropriate housing, and lacks the immediate prospect, means and ability of acquiring a home [2]. This definition assumes homelessness results from both a lack of affordable housing and an interplay between financial, cognitive, behavioral and physical challenges, or structural factors such as racism and discrimination. In Canada, for example, men who are chronically homeless face an estimated life expectancy of 43 years of age and women face a life expectancy of 53 years of age, compared to the average life expectancy of 80 years for men and 84 years for women in Canada. These premature and preventable deaths occur in marginalized populations and are associated with a large proportion of physical, mental health and substance use morbidity [3].

In preparing for our Delphi consensus method, we gained a new appreciation for the increasing diversity of homeless populations, including gender, age, ethnicity and types of indigenous populations. As such, we included a list of subpopulations we should address in the development of our clinical guidelines [4]. In 2014, for example, an estimated 235,000 people experienced homelessness in Canada, 27.3% of whom were women and 18.7% were youth, with a growing number of seniors [4,5]. Over-represented homeless populations included Indigenous Peoples (First Nations, Métis, and Inuit), people with disabilities, veterans, newly arrived refugees and other migrants, and gender diverse people [4]; while over one fifth of people with psychological or learning disabilities experience hidden homelessness [6]. As a consequence, we decided to include persons with lived experience of homelessness in our working group and as participants in the Delphi method [7].

Primary healthcare practitioners may benefit from structured training, support, and clinical guidelines to address the multimorbidity, advocacy and social needs of this population [8,9]. Engaging stakeholders, including people with lived experience, to prioritize needs and populations that will then be scientifically assessed using systematic reviews, may improve real-world trustworthiness and ultimately uptake of the final guidelines [10]. This approach has been implemented by the National Institute for Health and Care Excellence in the United Kingdom, which engages Citizen Panels to include the voices of lay members into clinical care guidelines

[11]. Our approach is informed by the methods outlined by the MuSE (Multi-Stakeholder Engagement Consortium) [12]. and the EQUATOR (Enhancing the QUAlity and Transparency Of health Research) group to develop health guidelines [13]. The objective of our Delphi consensus study was therefore to engage expert health professionals and people with lived experience in a priority setting consensus process for needs and populations and to ultimately guide the development of evidence-based clinical guidelines.

## Methods

### Study design

We used a modified Delphi approach using three phases, which are outlined below [8,14]. This method has been successfully used for priority setting for other marginalized populations [15–17].

Five members of the research team, which included two health professionals, family physicians and two persons with lived experience of homelessness, all with expertise in homeless health care, and a scientist with expertise in evaluating change, met in person in July 2017. The group reviewed literature and reports outlining the prevalence of disease and impact of homelessness as described in the introduction, as well as current approaches to delivering care for at risk populations. We pilot tested the survey with four expert health professionals and four lived experience participants. The working group modified questions to ensure multi-stakeholder clarity and acceptability. They developed an initial list of needs and at risk homeless health populations for experts to consider in recommending priorities for homeless health guidelines (see Delphi Round 1 in S1 Appendix).

### Ethics approval

We obtained ethics review and approval from the Bruyère Research Ethics Board (Ottawa, Ontario) (M16-17-012).

### Survey participants

We invited expert health professionals and persons with lived experience of homelessness from across Canada to participate in our Delphi. We used purposeful snowball sampling approaches among our homeless health networks. We purposefully selected participants to ensure the inclusion of a variety of local perspectives, including indigenous perspectives.

Canadian health professionals from family medicine, internal medicine, psychiatry and nursing were purposely identified using the following inclusion criteria: a) homeless health expertise and/or b) provincial or territorial diversity province and/or c) research in the area of homeless health. Every health professional was ultimately selected for their knowledge in homelessness. In addition, we decided to include people with lived experience of homelessness. E-mail invitations were sent to each expert to determine interest and to explain the time commitment involved in participating in the Delphi process.

Certain people were selected directly from our network, but the majority of participants were selected from 10 sites across Canada, shelters, food banks, and other community organizations. In these scenarios, one of our community outreach workers collaborated with local staff to select, invite and verbally deliver the survey to participants.

### Survey administration and analysis

The Delphi Consensus process included three surveys rounds. We administered three rounds of surveys using Survey Monkey from May 15, 2017 to November 15, 2017. Each survey was

live for 3–4 weeks and two reminders were sent. For people with lived experience, a research staff person interviewed the accepting participant and sent paper survey results by fax. The definition of the consensus was determined before the analysis of the round by the Delphi working group and in consultation with an epidemiologist. Investigators were blinded to the results during the data analysis. Follow-up was done through email and phone call or using local contacts with community partners. If a participant did not reply after three follow-up attempts over several weeks, they were removed from the next round.

### Round one

In Delphi round 1 (See S2 Appendix and S3 Appendix), we collected participants' characteristics such as practitioner specialty, age range and gender. We provided participants with a list of seven potential priority needs as well as a list of seven at-risk populations. The objective for all participants was to rank the highest priority needs considering value added (opportunity for a unique and relevant contribution), level of inequity (reduction of unfair and preventable health inequities) and burden of condition (number of people that may suffer from a disease or condition) [18]. They also ranked the most at risk populations from a list of seven previously identified populations. Microsoft Excel 2010 was used to run descriptive statistical analyses, including mean and standard deviation. Participants were also asked to list additional priority needs and at risk populations that were missing from the review generated lists. These were subsequently added to the list for round two (see Delphi Round 1 Survey, S1 Appendix).

### Round two

Round two of the Delphi consisted of two sections. Section one included the priority needs that ranked in the top 60% of participant ranking, including experts and people with lived experience. We would later report the experts and lived experience results separately but for our Delphi process we included both groups together. A mean rank (and standard deviation) was calculated for each need [19]. In section two, participants were asked to select their four highest priorities from a list of nine needs including those from the first round and an additional two based on participants' comments from round one. They were asked to consider the following criteria: added value, equity, and burden of disease.

### Round three

Round three of the Delphi survey included priority needs and at risk populations identified in the previous round. No free-text option was provided and no qualitative data was gathered during this round. People with lived experience had the option to answer step one and two surveys consecutively, and significant effort was made to follow up with them to include them in the third and final round of the survey.

The third and final round built consensus on the need and population ranking. We finalized rankings at a team meeting of experts working in the field of evidence-based homeless health. Electronic survey participants had the option to complete the survey using Survey Monkey or to request paper copies to facilitate completion. We analyzed the data using Microsoft Excel 2010.

### Results

We reached a 73% response rate among health professionals (114 invited and 84 completed the first round of the survey). It was difficult to estimate the response rate of people with lived experience of homelessness given the majority were approached by community volunteers

from the organization partners. Reasons why participants chose not to participate in all steps included leave of absence or sabbatical leave, clinical workload, or reasons not described. Please refer to Fig 1 for the number of participants in each round of the Delphi consensus process.

Table 1 outlines the demographic characteristics of all Delphi survey participants. In total, six Canadian provinces and ten urban centres were represented. The majority of participants came from Ontario (61.90% health professionals; 53.95% people with lived experience). Participants in both groups were well balanced in terms of gender and although all age groups were included in the survey, very few participants were younger than 25 years old whereas people with lived experience were on average slightly older than the health professional group. Approximately 80% of participants in both groups listed English as their first language. The sex, age, and first language distribution among people with lived experience and health professionals remained very similar across the three rounds. Among people with lived experience, 39 people experienced homelessness or being vulnerably housed for less than two years (51.32%) and eight participants reported 11+ years as their length of experiencing homelessness or being vulnerably housed (10.53%). Compared to the first round, people with lived experience who answered the third round of the Delphi were less likely to experience long term homelessness. Six people with lived experience also identified as health professionals.

Among health professionals, most worked as primary care providers, specialist physicians and registered nurses (n = 51, 60.71%), and others were researchers, public health experts, social workers, or community health advocates. Health professionals with different lengths of experience working with homeless populations participated in the Delphi survey; 24 (28.57%) of participants indicated 11+ years of experience in the field and 15 (17.86%) reported less than two years of experience.

Table 2 lists the prioritized and ranked needs from the Delphi consensus process by people with lived experience of homelessness and health professionals. Both groups prioritized, in the order of importance: facilitating access to housing, mental health and addiction care, care coordination/case management, and facilitating access to adequate income. There were few important differences in ranking between the two groups of participants. One difference was that health professionals ranked chronic disease management as the fifth priority while people with lived experience ranked nutrition and dietary support as their fifth priority. In terms of populations, both groups prioritized women, families, and children, Indigenous Peoples (First Nations, Métis, and Inuit), persons with acquired brain injury, intellectual, or physical disabilities, youth, and refugees and other migrants (see Table 3).

Table 4 outlines the relevance and importance of the needs and populations that were selected and that will be used to develop systematic reviews and then trustworthy clinical guidelines for practitioners to improve the health of people experiencing homelessness or who are vulnerably housed

## Discussion

Using a Delphi consensus method, guided by three criteria: value added, inequity, and burden of illness, we were able to identify and rank priority needs for people who are homeless or vulnerably housed in Canada. Early working group lists were more disease specific but health professionals and persons with lived experience of homelessness rankings rapidly shifted to more upstream social determinant of health needs such as income support and a shared consensus emerged between health professionals and people with lived experience.

The top four priority needs selected were: facilitating access to housing; providing mental health and addiction care; delivering care coordination and case management; and facilitating

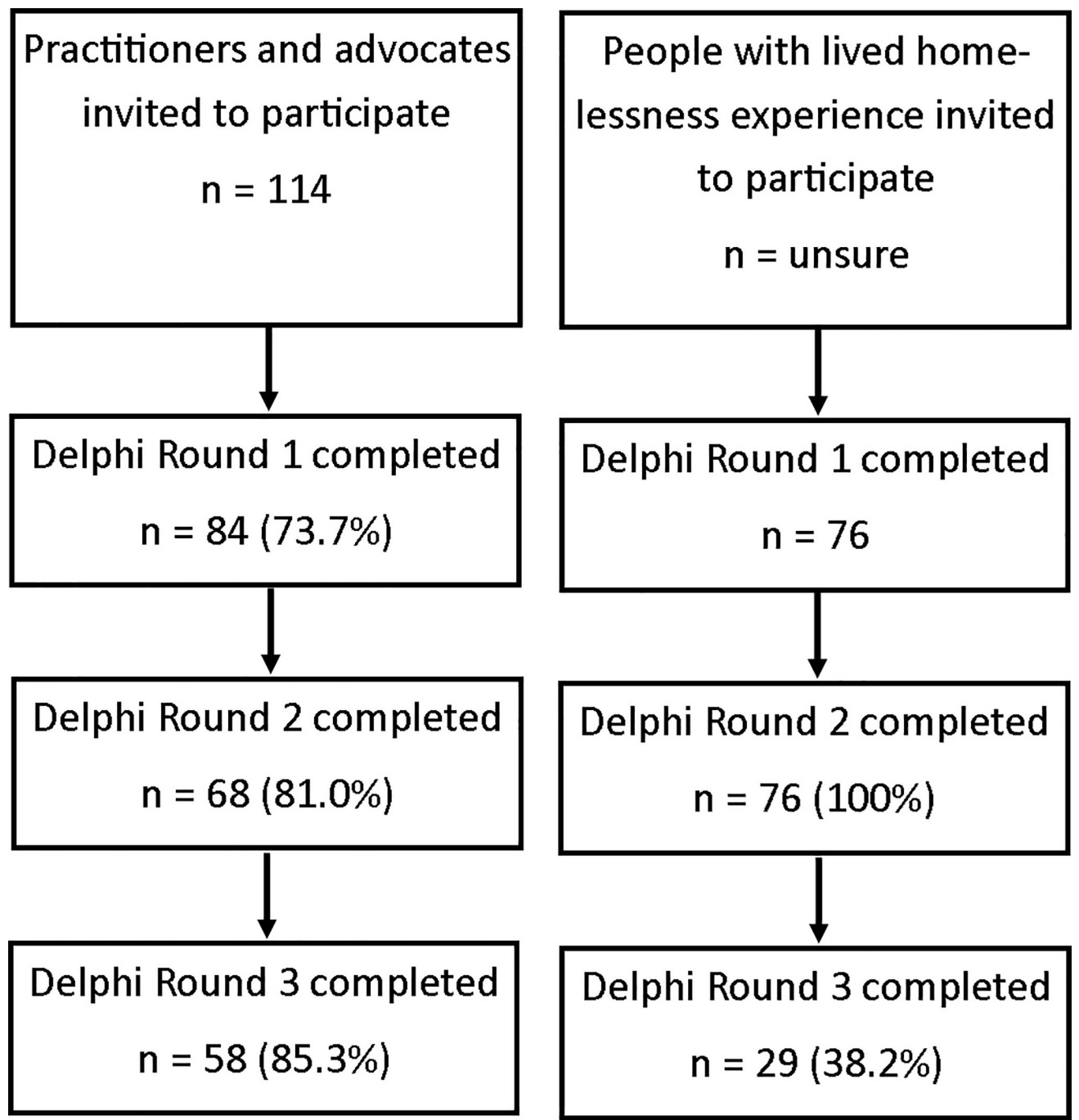

**Fig 1. Delphi survey participants sampling and response rate.**

access to adequate income. Access to housing, considered a basic human right [50], is a critical first step in implementing health and social care interventions for people experiencing homelessness and the prioritization of this need reflects the published research [51]. Prioritizing access to early housing have improved outcomes among people with serious mental illness [52,53], substance use disorders, veterans [26–28], and people experiencing homelessness in rural regions [54]. The provision of mental health and addiction care, selected by both groups

**Table 1. Demographic characteristics of Delphi survey participants for rounds 1, 2, and 3.**

| Characteristic | Health Professionals | | | People with Lived Experience | | |
|---|---|---|---|---|---|---|
| n (%) | Delphi Round 1 | Delphi Round 2 | Delphi Round 3 | Delphi Round 1 | Delphi Round 2 | Delphi Round 3 |
| | N = 84 | N = 66 | N = 58 | N = 76 | N = 76 | N = 29 |
| Age | | | | | | |
| < 30 | 8 (9.53) | 8 (12.12) | 7 (12.07) | 7 (9.21) | 6 (7.89) | 3 (10.34) |
| 31–40 | 23 (27.38) | 23 (34.85) | 18 (31.03) | 12 (15.79) | 13 (17.11) | 8 (27.59) |
| 41–50 | 27 (32.14) | 17 (25.76) | 17 (29.31) | 21 (27.63) | 21 (27.63) | 10 (34.48) |
| 51–60 | 15 (17.86) | 12 (18.18) | 12 (20.69) | 31 (40.79) | 30 (39.47) | 6 (20.69) |
| 61+ | 11 (13.10) | 6 (9.09) | 4 (6.9) | 5 (6.58) | 6 (7.89) | 2 (6.90) |
| Gender | | | | | | |
| Male | 36 (42.86) | 29 (43.94) | 23 (39.66) | 43 (56.58) | 43 (56.58) | 17 (58.62) |
| Female | 47 (55.96) | 36 (54.55) | 34 (58.62) | 33 (43.42) | 33 (43.42) | 12 (41.38) |
| Other | 1 (1.19) | 1 (1.52) | 1 (1.72) | - | - | - |
| Province | | | | | | |
| British Columbia | 3 (3.57) | 2 (3.03) | 1 (1.72) | 12 (15.79) | 12 (15.79) | - |
| Alberta | 5 (5.95) | 4 (6.06) | 4 (6.90) | 13 (17.11) | 11 (14.47) | 5 (17.24) |
| Manitoba | 1 (1.19) | - | - | - | 1 (1.32) | - |
| Ontario | 52 (61.90) | 46 (69.70) | 42 (72.42) | 41 (53.95) | 41 (53.95) | 10 (34.48) |
| Quebec | 16 (19.05) | 10 (15.15) | 8 (13.79) | 9 (11.84) | 10 (13.16) | 14 (48.28) |
| Nova Scotia | 3 (3.57) | 2 (3.03) | 2 (3.45) | - | - | - |
| Newfoundland and Labrador | 4 (4.76) | 2 (3.03) | 1 (1.72) | - | - | - |
| Missing | - | - | - | 1 (2.78) | 1 (1.32) | - |
| First language | | | | | | |
| English | 67 (79.76) | 56 (84.85) | 50 (86.21) | 61 (80.26) | 58 (76.32) | 15 (51.72) |
| French | 13 (15.48) | 7 (10.61) | 5 (8.62) | 4 (5.26) | 7 (9.21) | 1 (3.45) |
| Other[+] | 4 (4.76) | 3 (4.55) | 3 (5.17) | 3 (3.95) | 11 (14.47) | 13 (44.83) |
| Not reported | - | | - | 8 (10.53) | - | - |
| Profession | | | | | | |
| Primary care provider | 36 (42.86) | 33 (50.00) | 28 (48.28) | - | 1 (1.32) | 1 (3.57) |
| Specialist physician | 10 (11.90) | 8 (12.12) | 9 (15.52) | - | 1 (1.32) | - |
| Registered nurse | 5 (5.95) | 2 (3.03) | 3 (5.17) | 2 (2.63) | 2 (2.63) | 1 (3.57) |
| Public health expert | 5 (5.95) | 4 (6.06) | 4 (6.90) | 1 (2.78) | 1 (1.32) | 1 (3.57) |
| Social worker | 5 (5.95) | 2 (3.03) | 1 (1.72) | 3 (3.95) | 3 (3.95) | 1 (3.57) |
| Homelessness researcher | 16 (19.05) | 12 (18.18) | 10 (17.24) | 1 (2.78) | 1 (1.32) | 1 (3.57) |
| Community health advocate | 6 (7.14) | - | 1 (1.72) | 10 (13.16) | 2 (2.63) | - |
| Not applicable/missing | 11 (13.10) | 2 (3.03) | 2 (3.45) | 59 (77.63) | 8 (10.53) | - |
| Length of homelessness experience* | | | | | | |
| < 2 years | - | - | - | 39 (51.32) | 40 (52.63) | 16 (55.16) |
| 2–5 years | - | - | - | 17 (22.37) | 19 (25.00) | 10 (34.48) |
| 6–10 years | - | - | - | 12 (15.79) | 8 (10.53) | 2 (6.8.8) |
| 11+ years | - | - | - | 8 (10.53) | 9 (11.84) | 1 (3.48) |
| Length of involvement in homelessness research or programs | | | | | | |
| < 2 years | 15 (17.86) | 14 (21.21) | 14 (24.14) | 27 (35.53) | 27 (28.95) | 13 (44.83) |
| 2–5 years | 17 (20.24) | 8 (12.12) | 7 (12.07) | 13 (17.11) | 11 (14.47) | 9 (31.03) |
| 6–10 years | 18 (21.43) | 19 (28.79) | 15 (25.86) | 4 (5.26) | 6 (7.89) | 6 (20.69) |
| 11+ years | 24 (28.57) | 17 (25.76) | 17 (29.31) | 3 (3.95) | 4 (5.26) | - |
| Not applicable/missing | 10 (11.90) | 6 (9.09) | 5 (8.62) | 29 (38.16) | 28 (36.84) | 1 (3.45) |

**Table 2. Priority needs ranking.**

| Priority | People with Lived Experience | Health Professionals |
|---|---|---|
| 1 | Facilitating access to Housing | Facilitating access to Housing |
| 2 | Mental Health and Addiction Care/Trauma | Mental Health and Addiction Care/Trauma |
| 3 | Care coordination/Case management | Care coordination/Case management |
| 4 | Facilitating access to adequate income | Facilitating access to adequate income |
| 5 | Nutrition and dietary support | Chronic disease management |
| 6 | Chronic disease management (e.g. diabetes, smoking related lung disease) | HIV, Hepatitis B/C, TB, other infectious diseases |
| 7 | HIV, Hepatitis B/C, TB, other infectious diseases | Nutrition and Dietary support |
| 8 | Exposure related illnesses | End-of-life care |
| 9 | End-of-life care | Exposure related illness |

of Delphi participants, reflect the high prevalence of mental health conditions, and alcohol and substance use among people experiencing homelessness or who are vulnerably housed. Increasing our awareness of mental health difficulties and addictions among people experiencing homelessness or who are vulnerably housed is key to sustaining housing and community integration [55], and can prompt and inform research priorities.

Case management provides intentional person-centered support, assessment and planning in order to facilitate the delivery and uptake of health and social care services in a timely manner [56]. Effective case management can bridge care settings (i.e. inpatient or long-term care), care providers (i.e. informal caregivers, health specialists), and other resources (i.e. education, community services) to tailor an individualized care pathway, and has been shown to help individuals achieve housing stability [23]. The majority of people experiencing homelessness or who are vulnerably housed experience income insecurity [57]. Having identified access to income support as a priority by both Delphi groups participants supports evidence suggesting income as a critical determinant of health and well-being [58] and potential roles of care providers in mitigating consequences of income insecurity [37,59]. Delphi participants further identified specific populations that could benefit from targeted research to focus the guidelines specifically to their needs in addition to that of the population of people experiencing

**Table 3. Priority populations ranking.**

| Priority | People with Lived Experience | Health Professionals |
|---|---|---|
| 1 | Women, families and children | Indigenous (First Nations, Métis, Inuit) |
| 2 | People with acquired brain injury, intellectual, or physical disabilities | Women, families and children |
| 3 | Indigenous (First Nations, Métis, Inuit) | People with chronic homelessness |
| 4 | Refugees and migrants | Youth |
| 5 | Youth | Elderly |
| 6 | People with language barriers | People with acquired brain injury, intellectual, or physical disabilities |
| 7 | Elderly | Refugees and migrants |
| 8 | Victims of intimate partner violence / domestic abuse | People with diverse sexual orientations and/or gender diversity (LGBTQ) |
| 9 | People with diverse sexual orientations and/or gender diversity (LGBTQ) | Visible minorities |
| 10 | Visible minorities | People with language barriers |
| 11 | Veterans | Veterans |

**Table 4. Relevance and importance of high priority topics and populations.**

| Topic | | Importance |
|---|---|---|
| 1 | **Facilitating access to housing** | It is important to situate housing as a basic human right [20], irrespective of health and social service uptake [21]. |
| | | Initiatives that prioritize access to housing have demonstrated success among those with substance use disorders, veterans [22–24], and PLE from rural settings [25]. |
| 2 | **Providing mental health and addiction care** | A number of interventions have been developed for PLEs with mental illness and addictions [26] in Canada and internationally, including intensive case management [27], assertive community treatment [28], supportive and supported housing [29], housing first [30], critical time interventions [31], and harm reduction services such as managed alcohol programs [32], supervised injection sites and wet shelters [33]. |
| | | Such interventions are either not widely available or implemented with various degrees of fidelity to the evidence-based models [34]. Screening for mental health, addictions, and associated neurocognitive impairment and other disabilities among PLEs and building greater awareness of the range of supports available is essential to supporting PLEs in finding and keeping housing, addressing their mental health and substance use needs, and achieving community integration [35]. |
| 3 | **Delivering care coordination and case management** | Effective care coordination can bridge various care settings (i.e. inpatient or long-term care), potential participants (i.e. informal caregivers, health specialists), and other resources (i.e. education, community services) to create a unique care pathway tailored for the patient. Facilitating care coordination makes navigating complex health systems manageable for PLEs. |
| 4 | **Facilitating access to adequate income** | Case management programs for PLEs have included the need for obtaining adequate income at the centre of their support plans [36]. It is assumed that adequate income is a prerequisite for improving the health and increasing the likelihood of obtaining housing for PLEs. |
| | | The role of health providers in addressing income insecurity is increasingly recognized. Both the Canadian Medical Association and the College of Family Physicians of Canada have produced guidance documents for physicians on addressing income and other social determinants of health [37,38]. |
| | | Income intervention programs have been co-located with health care programs in the United Kingdom [39] and the United States [40] |
| **Population** | | **Importance** |
| 1 | **Indigenous people** | Indigenous people experience multiple risk factors for becoming homeless or vulnerably housed, such as low education level, insecure employment, and poor health [41]. Their distinct experience of being indigenous within a colonized country puts them at a structural and systematic disadvantage and at a significantly higher risk of homelessness or vulnerable housing [42]. |
| 2 | **Youth** | Youth who are PLEs have unique health needs as they experience high rates of substance use [43,44], frequent histories of exposure to domestic violence [45], and often resort to sex work to meet their basic needs once removed from the family setting [46]. |
| 3 | **Women, families, children** | Women, families, and children tend to be underrepresented among official homeless counts as they are more likely to be experiencing hidden homelessness and precarious housing compared to single men [47]. Women also have different paths into homelessness or vulnerable housing and suffer different sequelae than men [48]. |
| 4 | **People with acquired brain injury, intellectual, or physical disabilities** | Disability can lead to homelessness or vulnerable housing, as it is often accompanied by loss of income, social supports, and adequate housing [49]. |

homelessness. Participants prioritized: Indigenous Peoples (First Nation, Métis, and Inuit); youth; women, families, and children; and people with acquired brain injury, intellectual, or physical disabilities.

Indigenous Peoples in Canada include First Nations, Métis and Inuit populations. In urban settings, this population is over represented in Canada's homeless population. Although the prevalence varies by region, approximately 20–50% of those vulnerably housed or homeless are Indigenous [60]. Indigenous people experience multiple risk factors for becoming homeless or vulnerably housed, such as low education level, insecure employment and poor health [41], which are further exacerbated by structural and systematic barriers [42]. This finding sparked the development of an Indigenous researcher led approach for Indigenous people who are homeless or vulnerably housed [61].

Youth who are homeless or vulnerably housed are often difficult to identify and support due to their social situation and challenges relating to youth protection [62]. Precariously housed youth experience high rates of substance use [44,63], exposure to domestic violence [45], and often resort to sex work to meet their basic needs once removed from the family setting [46]. Women, families, and children are often underrepresented among official homeless counts [64] as they are more likely to be experiencing hidden homelessness compared to single men [6]. Women have different paths into various forms of homelessness, suffer different sequelae than men [48], and experience significant negative health consequences [65]. Disability is a significant feature among those who experience different forms of homelessness, particularly in terms of having acquired brain injuries [66], developmental disabilities, neurocognitive impairment, and musculo-skeletal injuries [67]. Disability, often accompanied by decrease or loss of income, social supports, and safe and secure housing [49] can become a precursor to homelessness.

## Strengths and limitations

The main strength of our study stems from the inclusion of people with lived experience of homelessness from across Canada. Our team collaborated with a diverse range of community organizations and sought need prioritization. Our study has a number of limitations. Repeatedly reaching persons with lived experience, most with no fixed address or contact numbers, was a significant challenge and meant accepting lower response rates over time. We did not include a substantive qualitative phase to the study and are unable to describe in detail the rationale for how participants prioritized the needs and populations. Finally, we are unable to conduct subgroup analysis of the needs ranking (e.g. for Indigenous Peoples) as the sample size of individual groups is too small.

## Conclusion

Our Delphi consensus method, with people with lived experience of homelessness and expert health professionals, uncovered priority needs for homeless populations. These needs sparked a series of systematic reviews and two distinct homeless health guidelines. Including people with lived experience provided a unique real world perspective on needs and marginalization. While medical conditions appeared on the initial list of needs, the voices of both health professionals and people with lived experience shifted the consensus to social determinants of health reflecting existing structural barriers. Providing mental health and addiction care was identified as the most important issue among both groups of respondents.

## Supporting information

**S1 Appendix. Results of ranking of needs and populations after Delphi rounds 1 and 2.**
(DOCX)

**S2 Appendix. Multi stakeholder Delphi consensus to Identify priorities for Canadian evidence-based guidelines to improve the health of homeless and vulnerably housed people.**
(PDF)

**S3 Appendix. Consensus de Delphi multipartite visant à déterminer les priorités des lignes directrices canadiennes fondées sur des données probantes pour améliorer la santé des sans-abri et des personnes vulnérables.**
(PDF)

## Author Contributions

**Conceptualization:** Esther S. Shoemaker, Claire E. Kendall, Anne Andermann, Gary Bloch, Tim Aubry, Peter Tugwell, Vicky Stergiopoulos, Kevin Pottie.

**Data curation:** Esther S. Shoemaker, Claire E. Kendall, Christine Mathew, Sarah Crispo, Vivian Welch, Anne Andermann, Sebastian Mott, Christine Lalonde, Gary Bloch, Alain Mayhew, Tim Aubry, Vicky Stergiopoulos.

**Formal analysis:** Esther S. Shoemaker, Claire E. Kendall, Christine Mathew, Sarah Crispo, Vivian Welch, Anne Andermann, Sebastian Mott, Christine Lalonde, Gary Bloch, Alain Mayhew, Vicky Stergiopoulos.

**Funding acquisition:** Claire E. Kendall, Anne Andermann, Gary Bloch, Tim Aubry, Peter Tugwell, Kevin Pottie.

**Investigation:** Esther S. Shoemaker.

**Methodology:** Esther S. Shoemaker, Claire E. Kendall, Vivian Welch, Anne Andermann, Kevin Pottie.

**Supervision:** Kevin Pottie.

**Writing – original draft:** Esther S. Shoemaker, Christine Mathew.

**Writing – review & editing:** Esther S. Shoemaker, Claire E. Kendall, Christine Mathew, Sarah Crispo, Vivian Welch, Anne Andermann, Sebastian Mott, Christine Lalonde, Gary Bloch, Alain Mayhew, Tim Aubry, Peter Tugwell, Vicky Stergiopoulos, Kevin Pottie.

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
