## [Decision Letter · Decision Letter 0]

28 Jan 2020

PONE-D-19-19733

Establishing need and population priorities to improve the health of homeless and vulnerably housed women, youth, and men: a Delphi consensus study

PLOS ONE

Dear Dr. Pottie,

Thank you for submitting your manuscript to PLOS ONE. After careful consideration, we feel that it has merit but does not fully meet PLOS ONE’s publication criteria as it currently stands. Therefore, we invite you to submit a revised version of the manuscript that addresses the points raised during the review process.

The Reviewers appreciated the study, though, both noted that the manuscript requires careful revision of the writing, especially in terms of clarity and fluency. I therefore suggest the Author to proceed with the reviewers' suggestions and, if necessary, to use a professional English language proofreading service. 

We would appreciate receiving your revised manuscript by Mar 13 2020 11:59PM. To enhance the reproducibility of your results, we recommend that if applicable you deposit your laboratory protocols in protocols.io, where a protocol can be assigned its own identifier (DOI) such that it can be cited independently in the future. For instructions see: http://journals.plos.org/plosone/s/submission-guidelines#loc-laboratory-protocols

We look forward to receiving your revised manuscript.

Kind regards,

Stefano Federici, Ph.D.

Academic Editor

PLOS ONE

Additional Editor Comments (if provided):

The Reviewers appreciated the study, though, both noted that the manuscript requires careful revision of the writing, especially in terms of clarity and fluency. I therefore suggest the Author to proceed with the reviewers' suggestions and, if necessary, to use a professional English language proofreading service.

Journal Requirements:

Reviewers' comments:

Reviewer's Responses to Questions

**Comments to the Author**

1. Is the manuscript technically sound, and do the data support the conclusions?

Reviewer #1: Yes

Reviewer #2: Partly

2. Has the statistical analysis been performed appropriately and rigorously? 

Reviewer #1: Yes

Reviewer #2: I Don't Know

3. Have the authors made all data underlying the findings in their manuscript fully available?

Reviewer #1: Yes

Reviewer #2: No

4. Is the manuscript presented in an intelligible fashion and written in standard English?

Reviewer #1: Yes

Reviewer #2: Yes

5. Review Comments to the Author

Reviewer #1: The Delphi technique is a tried-and-tested way for groups to build a consensus. Yet to date the method has seldom been used in homelessness. The value for Delphi panels in the context of evidence-use is that they can create an agreed view between researcher, professional and patient on what might work best, not a top-down ‘we know what research is best for you’ approach.

I was therefore delighted to see the technique used in this context to develop new policy and practice guidelines. The National Institute for Health and Care Excellence (NICE) in the UK uses Citizen Councils. While these are not strictly Delphi panels, they have a similar ethos: creating advice and guidelines that reflect a real-world consensus, not just rigidly following the scientific evidence from the lab.

The authors of the article took great care to ensure the inclusion of a diverse national group of people with lived homelessness experience. This is no small feat given the transient nature of the population.

Perhaps a little reframing in the introduction could have made it even clearer that the purpose of the work was to create rigorous and relevant guidelines to improve the care of people experiencing homelessness in Canada, building on a series of excellent systematic reviews. Applying this methodology will ensure the guidelines are as useful as possible in real life.

It is very positive that the authors are committed to continuing engagement efforts with an ever wider range of participants. Indeed, the fact that mental health and addiction care was identified as the most important issue among respondents suggests this will be important to highlight even further the structural causes of homelessness.

Reviewer #2: I have been asked to review "Establishing need and population priorities to improve the health of homeless and vulnerably housed women, youth, and men: a Delphi consensus study" for potential publication in PLoS One. This manuscript provides useful prioritization information in an important area of public health. I offer the following comments in order as they appear in the manuscript for the authors to consider:

- Abstract: Methods is difficult to follow grammatically. Could it be revised to "Initial needs were drafted from a literature review. To ensure accuracy and relevancy, we recruited 84 practitioners and 76 people with lived homelessness experience to participate in a 3-step Delphi consensus survey identifying and rating health and social priorities for homeless populations."

- Abstract: Methods would benefit from one line summarizing the criteria used to identifying individual needs and sub-populations as "high priority" or "most important".

- Abstract Findings would benefit from explicitly identifying the number of needs and sub-populations identified as "high priority".

- Introduction Lines 56-59: Could the authors also provide the life-expectancy figures for men and women who are not chronically homeless?

- Introduction Lines 59-61 make an unexpected jump to first-person language and discussion of the systematic reviews/Delphi conducted by this team. It reads as very out of place to me and could be removed, or at least moved elsewhere (e.g., to an appropriate place in the last paragraph of the introduction).

- Methods Lines 102-104 are difficult to follow grammatically.

- The Methods section would be easier to follow if sub-headers were used throughout. Per most reporting guidelines for empirical research, I recommend separate subsections for (1) participants (eligibility criteria and recruitment strategy), (2) data collection instruments (questionnaire items), (3) procedures (drafting of items, piloting, and running the process), and (4) analysis (how the authors analyzed the data from the Delphi). At the moment, the information in the methods section presents this information that seems "out of order" to me. For example, the authors discuss the Delphi survey, then they note the rankings were finalized during a meeting subsequent to the survey, and then they go back to information about conducting the Delphi survey (i.e., Survey Monkey or paper).

- The Methods section should include information on how many needs and populations were on the "initial list of needs and populations" (Line 122) and whether needs/populations were added or dropped in subsequent rounds.

- The Methods section should provide more information on the actual ranking task: were all needs and populations listed on one page? Did participants do a Q-sort task? Did they pick their top 10 (and then rank within it)? Please describe to the degree needed for the reader to replicate the ranking task. To this end, copies of the Delphi protocol and the survey for each round should be provided as an online supplement or included in an online repository (e.g., the Open Science Framework).

- In conjunction with more information on the ranking task, the Methods section also should provide more specific information on the analytic procedure used to analyze the ranking data for each round. For example, were rankings averaged across participants and then sorted? Please describe to the degree needed for the reader to replicate the analytic procedure.

- Methods Lines 140-141: The authors need to provide more details about the following: "We finalized rankings at a team meeting of experts working in the field of evidence-based homeless health." Who were the experts? How many were there at the meeting, and how were they chosen? When and for how long did the meeting take place? What information from the Delphi did they use at the meeting, and what was the specific procedure and analysis process for "finalizing" the rankings?

- Results Line 155: Figure 1 should include the response percentages as well as the N's.

- Results Lines 160-161 states "all age groups were included in the survey", though this is an artifact of the age categories created by the authors. The "<30" group particularly strikes me as one that should be split into more groups, given the growing public health concern about youth homelessness (including the statistics provided by the authors on youth homelessness).  

- Results Lines 177-186: These findings are difficult to interpret without the methodological context asked for above (i.e., the actual ranking task and the analytic procedure). The authors should provide more detail on how many items were considered in each round and provide a table with the actual ranking data on these items, including which and the number of items that do (and do not) meet their criteria for "most important" or "high priority". It is also not clear why only the needs are listed in a table: could similar information be provided for the sub-populations as well?

- Results Lines 187-189 are not results but rather are methods (see Lines 140-141). The authors here should report how many items came to this group, and the results of their discussions (e.g., how many items made the final list) according to the methodological details asked for above about the meeting.

- Discussion Line 193: I do not see a "Table 3" anywhere in the manuscript file.

- Discussion Lines 253-255: Please provide citations to support this claim.

- Discussion Lines 273-275: The authors refer to "the guideline development process", but have not provided the reader with any details about this. What guideline is being developed? Who is developing it? Who is involved? Who will use it once done?

6. PLOS authors have the option to publish the peer review history of their article (what does this mean?). If published, this will include your full peer review and any attached files.

Reviewer #1: Yes: Ligia Teixeira

Reviewer #2: Yes: Sean Grant

---

## [Author Response · Author response to Decision Letter 0]

26 Feb 2020

PONE-D-19-19733

Title: Establishing need and population priorities to improve the health of homeless and vulnerably housed women, youth, and men: a Delphi consensus study

Editor Comments: 

The Reviewers appreciated the study, though, both noted that the manuscript requires careful revision of the writing, especially in terms of clarity and fluency. I therefore suggest the Author to proceed with the reviewers' suggestions and, if necessary, to use a professional English language proofreading service.

Response: Thank you for this comment. We have thoroughly proofread and revised the manuscript and edited it for relevancy, clarity and fluency. 

Journal Requirements: When submitting your revision, we need you to address these additional requirements.

Response: Thank you, we have amended and used PLOS style requirements. 

Response: We have included the English and French version of the survey questionnaire as appendices. This questionnaire was developed based on a literature review and feedback from our Delphi team of researchers and persons with lived experience of homelessness. 

Reviewer #1: 

1. The Delphi technique is a tried-and-tested way for groups to build a consensus. Yet to date the method has seldom been used in homelessness. The value for Delphi panels in the context of evidence-use is that they can create an agreed view between researcher, professional and patient on what might work best, not a top-down ‘we know what research is best for you’ approach.

Response: Thank you. Yes, we sought a real world perspective that could lead to trustworthy guidelines. 

2. I was therefore delighted to see the technique used in this context to develop new policy and practice guidelines. The National Institute for Health and Care Excellence (NICE) in the UK uses Citizen Councils. While these are not strictly Delphi panels, they have a similar ethos: creating advice and guidelines that reflect a real-world consensus, not just rigidly following the scientific evidence from the lab.

Response: Thank you, this is a good point. Our patient and other stakeholder approach was guidelines by concepts from the MuSE project. (Petkovic D, MuSE Systematic Reviews BioMed Central 2020). The inclusion of homeless and vulnerably housed participants provided a real world perspective and consensus and provided leverage to build a network of users for the evidence. We agree with the reviewer that the mentioning of the NICE Citizen Councils would be useful and we have added a sentence into the last paragraph of our introduction: 

“This approach has been implemented by National Institute for Health and Care Excellence in the United Kingdom, which engages Citizen Panels to include the voices of lay members into clinical care guidelines. Our approach is informed by the methods outlined by the MuSE (Multi-Stakeholder Engagement Consortium) and the EQUATOR (Enhancing the QUAlity and Transparency Of health Research) group to develop health guidelines.” 

3. The authors of the article took great care to ensure the inclusion of a diverse national group of people with lived homelessness experience. This is no small feat given the transient nature of the population.

Response: Thank you

4. Perhaps a little reframing in the introduction could have made it even clearer that the purpose of the work was to create rigorous and relevant guidelines to improve the care of people experiencing homelessness in Canada, building on a series of excellent systematic reviews. Applying this methodology will ensure the guidelines are as useful as possible in real life.

Response: Thank you for this suggestion. We have reframed the introduction to stress the purpose of this study. The Delphi sought to inform systematic reviews and primary care focused evidence-based clinical guidelines.

5. It is very positive that the authors are committed to continuing engagement efforts with an ever wider range of participants. Indeed, the fact that mental health and addiction care was identified as the most important issue among respondents suggests this will be important to highlight even further the structural causes of homelessness.

Response: Thank you, we agree. We have highlighted this in the results and discussion section. 

Reviewer #2: 

1. I have been asked to review "Establishing need and population priorities to improve the health of homeless and vulnerably housed women, youth, and men: a Delphi consensus study" for potential publication in PLoS One. This manuscript provides useful prioritization information in an important area of public health. I offer the following comments in order as they appear in the manuscript for the authors to consider:

Response: Thank you, we appreciate your constructive comments.

2. Abstract: Methods is difficult to follow grammatically. Could it be revised to "Initial needs were drafted from a literature review (and community scholars). To ensure accuracy and relevancy, we recruited 84 practitioners and 76 people with lived homelessness experience to participate in a 3-step Delphi consensus survey identifying and rating health and social priorities for homeless populations." 

Response: Thank you. We have incorporated your advice and have rewritten the methods section of the abstract as follows: “We used a literature review and expert working group to produce a list of needs for homeless and vulnerably housed populations. We then followed a modified Delphi consensus method, asking expert clinicians, using electronic surveys, and persons with lived experience of homelessness, using oral surveys, to prioritise needs and at risk populations across Canada. Criteria for ranking included value added, inequities and burden of illness. We set ratings of ≥ 60% to determine consensus over three rounds of surveys.“

3. Abstract: Methods would benefit from one line summarizing the criteria used to identifying individual needs and sub-populations as "high priority" or "most important". 

Response: Please see our answer to the second comment. We have added the criteria used.

4. Abstract Findings would benefit from explicitly identifying the number of needs and sub-populations identified as "high priority". 

Response: Yes, we agree that this needed clarification and have listed the needs and populations identified as priority.

5. Introduction Lines 56-59: Could the authors also provide the life-expectancy figures for men and women who are not chronically homeless? 

Response:Thank you for this suggestion. We agree that these life expectancy statistics are necessary in order for the reader to understand the gravity of this issue. We have added these numbers to our introduction.

6. Introduction Lines 59-61 make an unexpected jump to first-person language and discussion of the systematic reviews/Delphi conducted by this team. It reads as very out of place to me and could be removed, or at least moved elsewhere (e.g., to an appropriate place in the last paragraph of the introduction).

Response: Thank you for this suggestion, we have edited the introduction, removed this statement for the first paragraph and have added it in a more appropriate place in the last paragraph.

7. Methods Lines 102-104 are difficult to follow grammatically.

Response: We have significantly rewritten the methods section.

8. The Methods section would be easier to follow if sub-headers were used throughout. Per most reporting guidelines for empirical research, I recommend separate subsections for (1) participants (eligibility criteria and recruitment strategy), (2) data collection instruments (questionnaire items), (3) procedures (drafting of items, piloting, and running the process), and (4) analysis (how the authors analyzed the data from the Delphi). At the moment, the information in the methods section presents this information that seems "out of order" to me. For example, the authors discuss the Delphi survey, then they note the rankings were finalized during a meeting subsequent to the survey, and then they go back to information a: bout conducting the Delphi survey (i.e., Survey Monkey or paper).

Response: We agree that a clear structure would be beneficial and have reorganized the methods section using subheadings.

9. The Methods section should include information on how many needs and populations were on the "initial list of needs and populations" (Line 122) and whether needs/populations were added or dropped in subsequent rounds.

Response: We agree that this information would be helpful for the reader and have listed the numbers following the more detailed description of each of the Delphi rounds.

10. The Methods section should provide more information on the actual ranking task: were all needs and populations listed on one page? Did participants do a Q-sort task? Did they pick their top 10 (and then rank within it)? Please describe to the degree needed for the reader to replicate the ranking task. To this end, copies of the Delphi protocol and the survey for each round should be provided as an online supplement or included in an online repository (e.g., the Open Science Framework).

Response: As described in the response to comment 9, we have added significant detail on the ranking to the methods section and have also added the first round of our survey tool as an appendix.

11. In conjunction with more information on the ranking task, the Methods section also should provide more specific information on the analytic procedure used to analyze the ranking data for each round. For example, were rankings averaged across participants and then sorted? Please describe to the degree needed for the reader to replicate the analytic procedure.

Response: We have added these details to the methods section. The analyses are described within the descriptions for each of the survey rounds.

12. Methods Lines 140-141: The authors need to provide more details about the following: "We finalized rankings at a team meeting of experts working in the field of evidence-based homeless health." Who were the experts? How many were there at the meeting, and how were they chosen? When and for how long did the meeting take place? What information from the Delphi did they use at the meeting, and what was the specific procedure and analysis process for "finalizing" the rankings?

Response: The final priorities were selected directly based on the results of the delphi. The expert meeting that followed was intended to begin the formation of working groups for each of the needs identified. We have decided that a description is not beneficial to this specific study and have therefore removed it from the methods section.

13. Results Line 155: Figure 1 should include the response percentages as well as the N's.- ok

Response: Thank you for the suggestion, we have included the percentages after the n’s.

14. Results Lines 160-161 states "all age groups were included in the survey", though this is an artifact of the age categories created by the authors. The "<30" group particularly strikes me as one that should be split into more groups, given the growing public health concern about youth homelessness (including the statistics provided by the authors on youth homelessness). 

Response:We agree that the youth are a key group as has been identified by the participants of the delphi. However, we cannot make the <30 age group smaller for risk of exposing single participants as the overall number of participants in that group is too small. We have modified our statement in the results section to read: “Participants in both groups were well balanced in terms of gender and even though all age groups were included in the survey, very few participants were younger than 25 years old with but people with lived experience being slightly older than the health professional group.”

15. Results Lines 177-186: These findings are difficult to interpret without the methodological context asked for above (i.e., the actual ranking task and the analytic procedure). The authors should provide more detail on how many items were considered in each round and provide a table with the actual ranking data on these items, including which and the number of items that do (and do not) meet their criteria for "most important" or "high priority". It is also not clear why only the needs are listed in a table: could similar information be provided for the sub-populations as well?

Response: As described in response to comments 9 and 10, we have rewritten the methods section to include these details. We have also added the priority populations ranking as a table 3, and the scores for the rankings of needs and populations for each of the rounds as tables in Appendix A. We did not have subpopulation needs ranking, this was not possible and would have required a large participant with good representation of the varous group. 

16. Results Lines 187-189 are not results but rather are methods (see Lines 140-141). The authors here should report how many items came to this group, and the results of their discussions (e.g., how many items made the final list) according to the methodological details asked for above about the meeting.

Response: Revised as described above.

17. Discussion Line 193: I do not see a "Table 3" anywhere in the manuscript file. ???

Response: It is included now as table 4.

18. Discussion Lines 253-255: Please provide citations to support this claim.--???

Response: We have added the following reference: “Anderson JT, Collins D. Prevalence and Causes of Urban Homelessness Among Indigenous Peoples: A Three-Country Scoping Review. Housing Studies; 2014;29(7):959-976.”

19. Discussion Lines 273-275: The authors refer to "the guideline development process", but have not provided the reader with any details about this. What guideline is being developed? Who is developing it? Who is involved? Who will use it once done?

Response: We have now included these details in the introduction and have summarized them in the conclusion. This Delphi method provided the priority needs and we used systematic reviews to determine effective and cost effective interventions using the GRADE Working Group methods approach. 

---

## [Decision Letter · Decision Letter 1]

1 Apr 2020

Establishing need and population priorities to improve the health of homeless and vulnerably housed women, youth, and men: a Delphi consensus study

PONE-D-19-19733R1

Dear Dr. Pottie,

We are pleased to inform you that your manuscript has been judged scientifically suitable for publication and will be formally accepted for publication once it complies with all outstanding technical requirements.

With kind regards,

Stefano Federici, Ph.D.

Academic Editor

PLOS ONE

Additional Editor Comments (optional):

Reviewers' comments:

Reviewer's Responses to Questions

**Comments to the Author**

1. If the authors have adequately addressed your comments raised in a previous round of review and you feel that this manuscript is now acceptable for publication, you may indicate that here to bypass the “Comments to the Author” section, enter your conflict of interest statement in the “Confidential to Editor” section, and submit your "Accept" recommendation.

Reviewer #2: All comments have been addressed

2. Is the manuscript technically sound, and do the data support the conclusions?

Reviewer #2: Yes

3. Has the statistical analysis been performed appropriately and rigorously? 

Reviewer #2: Yes

4. Have the authors made all data underlying the findings in their manuscript fully available?

Reviewer #2: No

5. Is the manuscript presented in an intelligible fashion and written in standard English?

Reviewer #2: Yes

6. Review Comments to the Author

Reviewer #2: (No Response)

7. PLOS authors have the option to publish the peer review history of their article (what does this mean?). If published, this will include your full peer review and any attached files.

Reviewer #2: Yes: Sean Grant

---

## [Editor Report · Acceptance letter]

6 Apr 2020

PONE-D-19-19733R1 

Establishing need and population priorities to improve the health of homeless and vulnerably housed women, youth, and men: a Delphi consensus study 

Dear Dr. Pottie:

I am pleased to inform you that your manuscript has been deemed suitable for publication in PLOS ONE. Congratulations! Your manuscript is now with our production department. 

With kind regards,

on behalf of

Prof. Stefano Federici 

Academic Editor

PLOS ONE